**Data Availability Statement:** All data collated as part of this study, as well as the analytical code used to generate the analyses reported in the Main

# Factors associated with variation in single-dose albendazole pharmacokinetics: A systematic review and modelling analysis

**Charles Whittaker**[1,2]*, **Cédric B. Chesnais**[3], **Sébastien D. S. Pion**[3], **Joseph Kamgno**[4], **Martin Walker**[2,5], **Maria-Gloria Basáñez**[1,2‡], **Michel Boussinesq**[3‡]

**1** MRC Centre for Global Infectious Disease Analysis, Department of Infectious Disease Epidemiology, School of Public Health, Imperial College London, London, United Kingdom, **2** London Centre for Neglected Tropical Disease Research, Department of Infectious Disease Epidemiology, School of Public Health, Imperial College London, London, United Kingdom, **3** Recherches Translationnelles sur le VIH et les Maladies Infectieuses (TransVIHMI), University of Montpellier, Institut de Recherche pour le Développement (IRD), Institut National de la Santé et de la Recherche Médicale (INSERM), Montpellier, France, **4** Centre for Research on Filariasis & other Tropical Diseases, and Faculty of Medicine and Biomedical Sciences, University of Yaoundé I, Yaoundé, Cameroon, **5** Department of Pathobiology and Population Sciences, Royal Veterinary College, Hatfield, United Kingdom

‡ These authors are joint senior authors on this work.
* charles.whittaker16@imperial.ac.uk

## Abstract

### Background

Albendazole is an orally administered anti-parasitic medication with widespread usage in a variety of both programmatic and clinical contexts. Previous work has shown that the drug's pharmacologically active metabolite, albendazole sulfoxide, is characterised by substantial inter-individual pharmacokinetic variation. This variation might have implications for the efficacy of albendazole treatment, but current understanding of the factors associated with this variation remains incomplete.

### Methodology/Principal findings

We carried out a systematic review to identify references containing temporally disaggregated data on the plasma concentration of albendazole and/or (its pharmacologically-active metabolite) albendazole sulfoxide following a single oral dose. These data were then integrated into a mathematical modelling framework to infer albendazole sulfoxide pharmacokinetic parameters and relate them to characteristics of the groups being treated. These characteristics included age, weight, sex, dosage, infection status, and whether patients had received a fatty meal prior to treatment or other drugs alongside albendazole. Our results highlight a number of factors systematically associated with albendazole sulfoxide pharmacokinetic variation including age, existing parasitic infection and receipt of a fatty meal. Age was significantly associated with variation in albendazole sulfoxide systemic availability and peak plasma concentration achieved; as well as the clearance rate (related to the half-life) after adjusting for variation in dosage due to differences in body weight between children and adults. Receipt of a fatty meal prior to treatment was associated with

Text and Supplementary File can be found at the following link: https://github.com/cwhittaker1000/albendazole_pk.

**Funding:** C.W. acknowledges funding from the Wellcome Trust (224190/Z/21/Z). C.W. and M.G.B. acknowledge funding from the Medical Research Council (MRC) Centre for Global Infectious Disease Analysis (MR/R015600/1), jointly funded by the UK MRC and the UK Foreign, Commonwealth & Development Office (FCDO), under the MRC/FCDO Concordat agreement and is also part of the European and Developing Countries Clinical Trials Partnership (EDCTP2) programme supported by the European Union. The funders had no role in study design, data collection and analysis, decision to publish, or preparation of the manuscript.

**Competing interests:** The authors have declared that no competing interests exist.

increased albendazole sulfoxide systemic availability (and by extension, peak plasma concentration and total albendazole sulfoxide exposure following the dose). Parasitic infection (particularly echinococcosis) was associated with altered pharmacokinetic parameters, with infected populations displaying distinct characteristics to uninfected ones.

## Conclusions/Significance

These results highlight the extensive inter-individual variation that characterises albendazole sulfoxide pharmacokinetics and provide insight into some of the factors associated with this variation.

### Author summary

Albendazole is a broad-spectrum anti-parasitic medication widely used in the treatment of a variety of parasitic worm infections. Previous studies have demonstrated marked variation in the pharmacokinetics of albendazole (and its pharmacologically active metabolite albendazole sulfoxide), leading to substantial inter-individual variability in blood plasma concentrations of albendazole sulfoxide following an oral dose of albendazole. This variation is thought to have consequences for treatment success but our understanding of the factors driving this variation remains incomplete. In this study, we carried out a systematic review to identify references with data on albendazole and albendazole sulfoxide concentrations in plasma following a single oral dose. We then fitted a mathematical model of albendazole sulfoxide pharmacokinetics to these data to infer key pharmacokinetic parameters and relate them to characteristics of the groups being treated. We found that: 1) receipt of a fatty meal prior to treatment was associated with increased albendazole sulfoxide systemic availability; 2) the half-life of albendazole sulfoxide varied significantly with age, and 3) both echinococcosis and neurocysticercosis were associated with altered albendazole sulfoxide pharmacokinetic profiles compared to healthy individuals. Our work provides insight into some of the factors systematically associated with variation in albendazole sulfoxide pharmacokinetics.

## Introduction

Albendazole is a broad-spectrum medication used widely in the treatment of a variety of parasitic worm infections. This includes usage in a clinical context, where multiple-dose regimens are used to treat infections with the larval stages of *Taenia solium* ((neuro-)cysticercosis) [1] or of *Echinococcus* spp. (principally cystic and alveolar echinococcosis due to, respectively, *E. granulosus* and *E. multilocularis*) [2]. It has also been used extensively in neglected tropical disease (NTD) programmatic contexts, for which albendazole is being/has been delivered (as a single-dose treatment) to communities in mass drug administration (MDA) campaigns against soil-transmitted helminthiases [3] (STHs, due to *Ascaris lumbricoides* (roundworm), *Trichuris trichiura* (whipworm), and *Necator americanus* and/or *Ancylostoma duodenale* (hookworm)), and delivered alone [4] or alongside ivermectin and/or diethylcarbamazine [5,6]) against lymphatic filariasis [7] (LF, due to *Wuchereria bancrofti* or *Brugia malayi*). In addition, albendazole has also been offered to individuals with loiasis [8,9], frequently those whose *Loa loa* microfilarial densities are high enough to preclude safe treatment with microfilaricidal anthelmintics [10,11] (such as diethylcarbamazine or ivermectin [12]).

Whilst the therapeutic efficacy of albendazole has been established for a wide array of helminth parasites, the pharmacokinetics of the drug's pharmacologically-active metabolite, albendazole sulfoxide, are characterised by extensive inter- and intra-individual variation. This variation has been consistently observed across a wide range of studies (see [13] for a review and its implications for treatment). It is typically attributed to the drug's limited solubility in the gastrointestinal tract (thought to be driven, in part, by inter-individual variability in gastric pH and intestinal metabolism, which can significantly impact the drug's absorption and bioavailability [14–16]) as well as extensive first-pass metabolism by the liver (responsible for rapid conversion of albendazole to albendazole sulfoxide).

Previous work has highlighted a relationship between higher albendazole sulfoxide blood plasma (henceforth referred to as plasma) levels and increased antiparasitic efficacy in patients with neurocysticercosis [17]. Similarly, higher dosages of albendazole have been associated with higher efficacy against hookworm [18], as have utilisation of triple-dose regimens over single-dose ones [19]. In this context, substantial inter-individual pharmacokinetic variability might contribute to the failure of cure observed in some treated patients. Indeed, across both individual treatment contexts in healthcare facilities [13,20,21] and field studies [22], highly variable impact of the drug on clearing treated infections has been observed. However, some of these observations might also be driven by other factors, such as variation between settings in the prevalence of different STH species which are thought to show variable responses to albendazole treatment [23].

A number of factors are thought to underlie this pharmacokinetic variation. Several studies have examined the influence of different drivers, including sex [24], co-administered drugs [14,25], delivery of albendazole alongside a fatty meal [26–29] and infection status [30,31] on the pharmacokinetic profile of albendazole sulfoxide. However, these studies typically only analyse a single factor, and therefore a systematic understanding of the respective comparative impact of different factors on albendazole's (and particularly albendazole sulfoxide's) pharmacokinetics remains outstanding. Given albendazole's widespread usage in NTD programmatic contexts, insight into mechanisms that influence the pharmacokinetic profile of albendazole sulfoxide could have significant public health relevance.

Motivated by this, we conducted a systematic review of the literature to identify references containing temporally disaggregated information on albendazole sulfoxide (and albendazole where available) concentrations in plasma following treatment with a single oral dose (the typical regimen used in NTD programmatic contexts). To these data, we fitted a mathematical model of albendazole and albendazole sulfoxide's pharmacokinetics following receipt of the dose that captures key phenomena associated with the drug's metabolism. These include albendazole's extensive first-pass metabolism [32] and its established low systemic availability [15]. We fitted this model to data collated as part of the systematic review to infer key pharmacokinetic parameters, including albendazole sulfoxide systemic availability, albendazole sulfoxide half-life, peak albendazole sulfoxide concentration in the plasma ($C_{Max}$) and the total exposure to albendazole sulfoxide across time (commonly described as the area under the curve or $AUC$). We then related these parameter estimates to characteristics of the patient groups being treated and the treatment regimen received.

## Methods

### Systematic review of albendazole pharmacokinetics literature

Web of Science and PubMed databases were searched on $22^{nd}$ July 2022 with no limitations on date range using the keywords "albendazole" AND (treatment* OR dose* OR pharma* OR "half-life" OR "half life") in order to identify references containing temporally disaggregated

data detailing the concentration of albendazole and/or albendazole sulfoxide in plasma following treatment with a single dose of the drug. A total of 6,855 unique records were identified through this search process, with 246 records retained for full text evaluation following Title and Abstract screening (Fig 1). Exclusion criteria comprised studies lacking the required information on plasma concentration levels over time; those that had been carried out *in vitro* or in non-human subjects, or were not in English. Following this, a total of 36 references were included, yielding 113 time-series describing the evolution of plasma concentrations of albendazole and/or albendazole sulfoxide following treatment with a single dose. Of these, 19 time-series contained information on both albendazole and albendazole sulfoxide levels, and 94 contained information on albendazole sulfoxide levels only. A total of 105 of these time-series

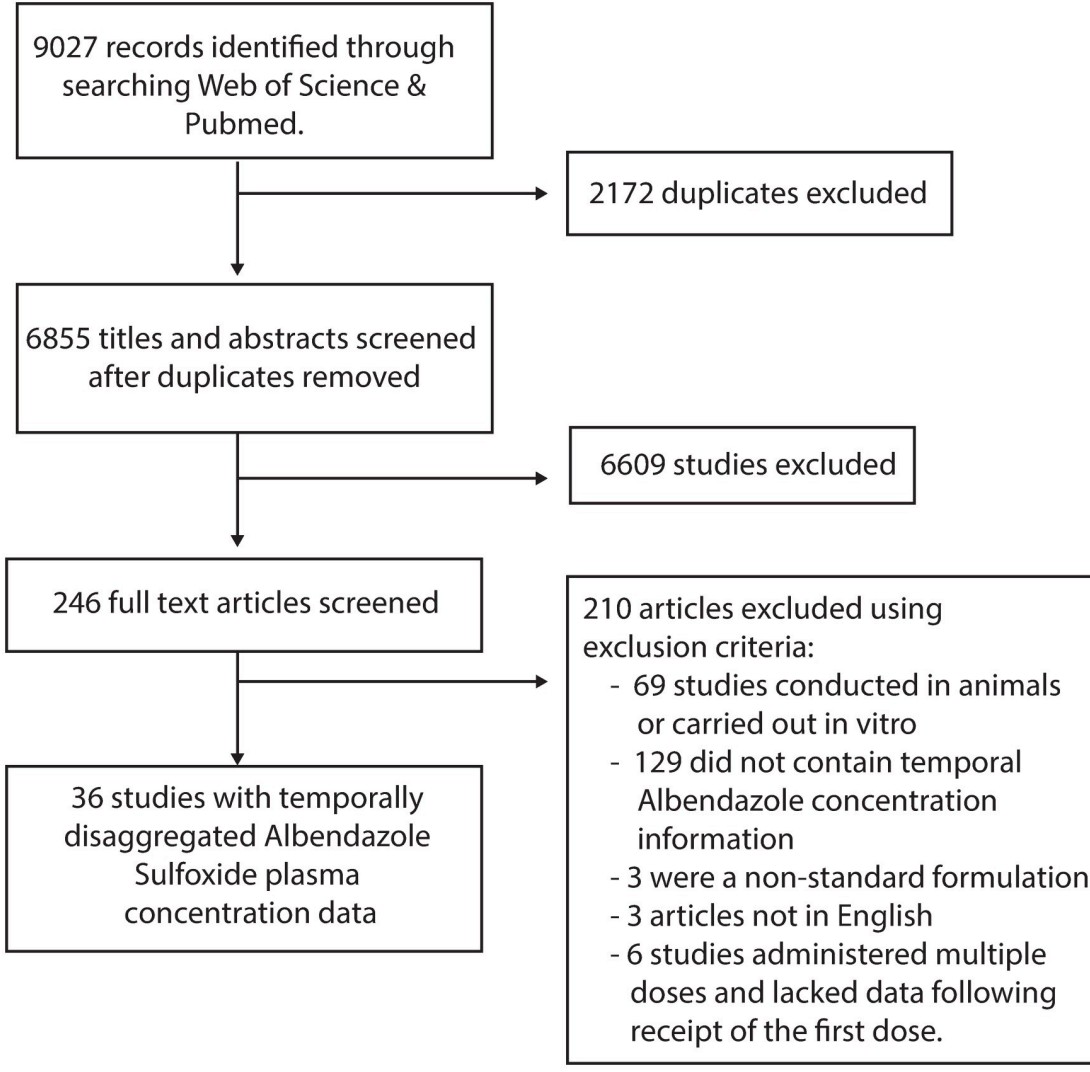

**Fig 1. PRISMA diagram illustrating the systematic review workflow.** Web of Science and PubMed were searched on 22[nd] July 2022 using the keywords albendazole AND (treatment* OR dose* OR pharma* OR "half-life" OR "half life"). This produced a total of 6,855 results after duplicate removal, of which 246 were retained for full text screening. A total of 210 of the retained articles were subsequently excluded based on pre-defined exclusion criteria (see Systematic review of albendazole pharmacokinetics literature, in Methods), yielding 36 studies containing temporally disaggregated data on albendazole sulfoxide plasma concentrations following treatment with a single dose of albendazole. These 36 references contained a total of 113 time-series measuring albendazole and/or albendazole sulfoxide blood plasma concentrations over time in different groups.

were studies in which only plasma albendazole sulfoxide concentrations following a single oral dose of albendazole were presented. In eight studies, measured plasma albendazole sulfoxide concentrations were presented that spanned a course of multiple doses, also containing information on albendazole plasma concentration levels immediately following the first dose. For these eight time-series, we extracted data following receipt of the first dose up until receipt of the second dose. For each time-series, we also extracted the data describing evolution of albendazole/albendazole sulfoxide plasma concentrations over time, as well as an array of metadata. These included characteristics of the treatment regimen (dose, fasting state, co-administered drugs), as well as information and metadata on the patients receiving treatment (sex, age, infection status and weight). In the majority of instances, presented data were reported for a group of patients rather than individuals. In these instances, group averages for factors such as age, weight, etc., were extracted. A full list of these references, as well as further information about each study and how the data were extracted are available in **Text A in** S1 File.

## Mathematical model construction and fitting

We developed a model describing the evolution of albendazole and albendazole sulfoxide concentrations in plasma following receipt of a single oral dose of the drug, based on a series of linked ordinary differential equations (ODEs) of albendazole and albendazole sulfoxide (Fig 2). The model incorporates a number of relevant pharmacokinetic phenomena, including the drug's well-established, limited systemic availability (thought to be a product of its poor solubility along the gastrointestinal tract [15]) and the extensive first-pass metabolism of albendazole to albendazole sulfoxide known to occur in the liver [32]. This model was fitted individually to each of the 113 collated time-series within a Bayesian framework, utilising an adaptive Metropolis-Hastings based Markov Chain Monte Carlo (MCMC) sampling scheme for parameter inference. Uninformative priors were used for each of the parameters being

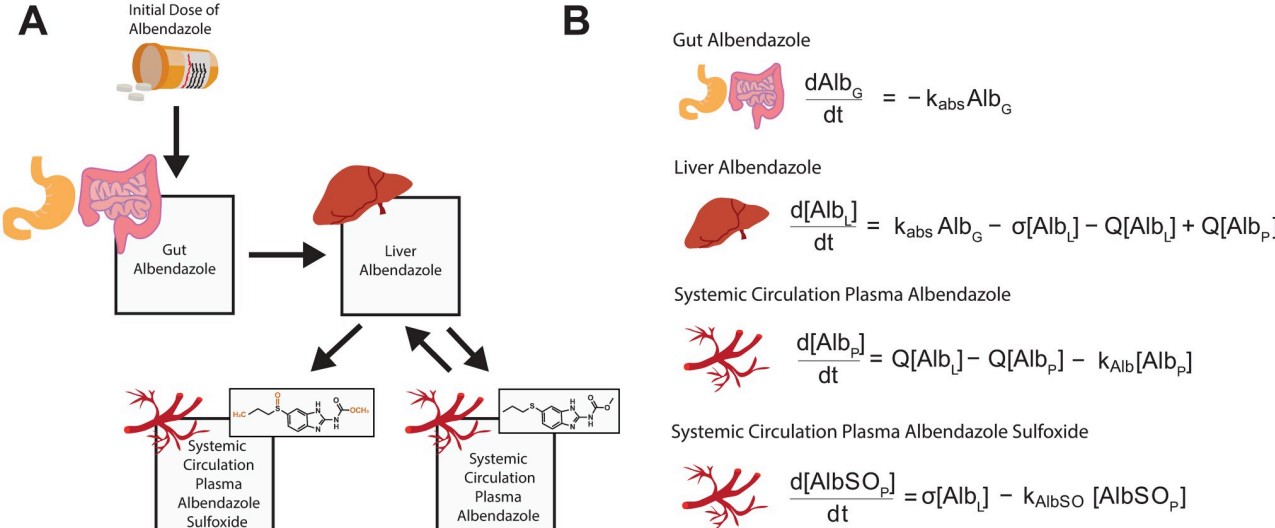

**Fig 2. Schematic representation of the model describing albendazole and albendazole sulfoxide pharmacokinetics.** A compartmental model consisting of a series of linked ordinary differential equations (ODEs) was developed to simulate the pharmacokinetics of albendazole and its pharmacodynamically-active metabolite, albendazole sulfoxide, in plasma following a single oral dose. **(A)** Schematic representation illustrating the model structure and the way in which the different compartments are linked. **(B)** Overview of the ODEs governing the model, representing the amount of albendazole in the gut (G) and concentration of the drug (or its metabolite albendazole sulfoxide) in the liver (L) and systemic circulation, plasma (P). The received dose is further scaled by a systemic availability parameter not shown here (see **Text B in** S1 File for further details and full description of the model equations).

subsequently related to the collected metadata. For each dataset, a total of 25,000 iterations were run, with the first 5,000 discarded as burn-in, leaving 20,000 samples available for parameter inference. The median value based on these (posterior) samples was then used as an input into the multiple linear regression analysis described below. Further information on the exact formulation of the model and the fitting process is available in **Text B in** S1 File.

### Regression linking pharmacokinetic properties to patients' characteristics

From the 113 fitted time-series, we extracted estimates of key pharmacokinetic parameters and regressed them onto the collected metadata (describing aspects of the patient group and treatment regimen received) to assess the influence of various factors on variation in albendazole and albendazole sulfoxide's pharmacokinetics. The pharmacokinetic parameters were $k_{Albso}$ (the clearance rate of albendazole sulfoxide, the reciprocal of which i.e. $1/k_{Albso}$ is the albendazole sulfoxide half-life), the systemic availability of albendazole sulfoxide (the proportion of administered albendazole absorbed from the gut and detected in the plasma as albendazole sulfoxide), $C_{Max}$ (the peak concentration of albendazole sulfoxide in plasma), and $AUC$ ("area under the curve", reflecting the total exposure to albendazole sulfoxide after administration of the dose, calculated over a time-period of 50 hours). $k_{Albso}$ and the systemic availability of albendazole sulfoxide are model parameters directly estimated during the fitting process outlined above, and so for each time-series, the median parameter estimate from each time-series was used in the regression. For $C_{Max}$ and $AUC$, in order to control for differences in dosages between studies (which would directly impact estimates of these two quantities), we used the fitted model (and median parameter estimates) for each time-series to simulate and generate a hypothetical pharmacokinetic curve assuming a standardised dose of 400 mg. We then calculated $C_{Max}$ and $AUC$ from this hypothetical pharmacokinetic curve to give estimates of the two parameters standardised by the dose received. We subsequently refer to these quantities as $C_{Max400}$ and $AUC_{400}$.

## Results

### Systematic review results and study characteristics

A total of 36 references containing 113 time-series detailing the concentration of albendazole and/or albendazole sulfoxide in plasma following treatment with a single dose of albendazole were identified [14,25–30,33–61]. Forty-four time-series were data for a single individual and 69 time-series described average plasma concentrations through time for a group of individuals (mean group size = 13.4, interquartile range = 6–18), with the data comprising a total number of 967 individuals who had received a single dose of albendazole. Of the 113 time-series identified, information on the sex of participants was available for 88 time-series (37 from male participants, 45 including a mixture of males and females, and 6 from female participants), with information allowing calculation of mean age and weight available for 94 and 97 time-series respectively. A total of 23 time-series were from children under the age of 18 years. Information on whether treatment was taken with a fatty meal was available for 91 time-series (39 received a fatty meal, the remainder did not), whilst infection status was available for 111 time-series (58 were from uninfected patient groups; 16 were from individuals with neurocysticercosis; 14 from individuals with echinococcosis; 11 from individuals with soil-transmitted helminth infections (either whipworm or hookworm); 7 from individuals with onchocerciasis; 3 from individuals with LF, and 2 from individuals with giardiasis). The median dose received was 400 mg (range 200 mg–2,205 mg). Co-administered drugs included ivermectin (n = 9), diethylcarbamazine (DEC, n = 9), praziquantel (n = 4), ritonavir (n = 2), dexamethasone (n = 2), amoxicillin (n = 1), gentamycin (n = 1), metronidazole (n = 1), ceftriaxone (n = 1), levamisole (n = 1) and oxantel pamoate (n = 1). **Table A in** S1 File provides full details of each included study and time-series.

## Pharmacokinetic modelling of albendazole and albendazole sulfoxide

To each of these collated time-series, we fitted a model describing the pharmacokinetics of albendazole and albendazole sulfoxide concentrations in plasma following receipt of a single oral dose and before a second dose in the case of treatment regimens using multiple doses (see Fig 2 for model structure and formulation). This model was fitted individually to each time-series within a Bayesian MCMC-based framework (see **Fig A in** S1 File for individual model fitting results for each time-series). Our results highlighted significant variation in model estimates of key pharmacokinetic parameters including $k_{Albso}$, the systematic availability of albendazole sulfoxide, $C_{Max400}$ (peak modelled concentration of albendazole sulfoxide in plasma following receipt of a hypothetical 400 mg dose) and $AUC_{400}$ (total modelled exposure to albendazole sulfoxide following receipt of a hypothetical 400 mg dose of albendazole). Stratifying the modelled pharmacokinetic profiles by various characteristics of the patient groups suggested possible systematic pharmacokinetic differences associated with patient- and treatment regimen-related factors, although also extensive between-study variation in pharmacokinetics and plasma concentration of albendazole sulfoxide over time following the dose (Fig 3).

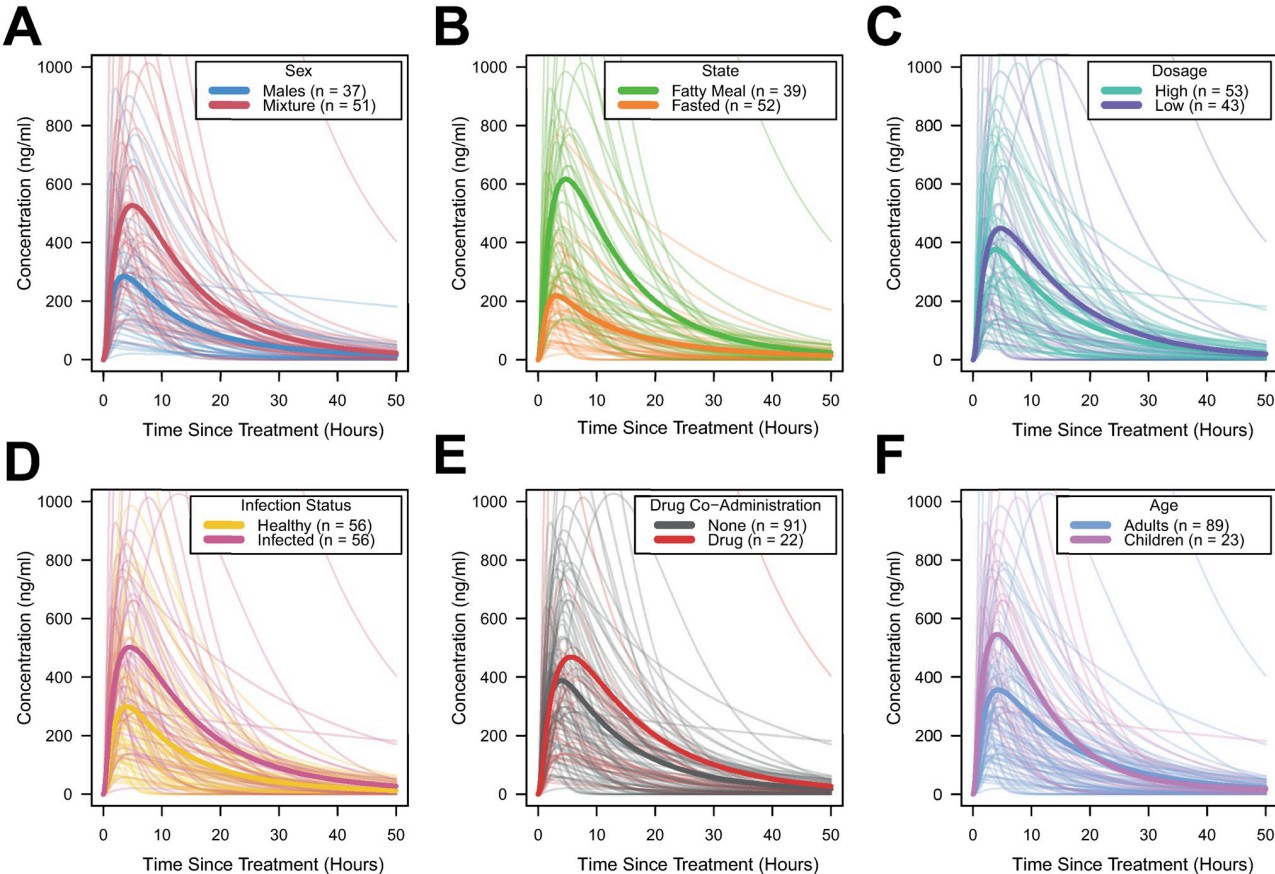

**Fig 3. Albendazole sulfoxide pharmacokinetic variability, stratified by patient and dosage features.** In all panels displayed above, each pale line represents the fitted model output for a single time-series, with the darker lines representing the average of the time-series for a given category. Factors explored were **(A)** Sex; **(B)** Feeding Status (according to whether groups had received the single dose of albendazole alongside a fatty meal or not); **(C)** Dose (with time-series crudely categorised into high/low strata based on whether the dose was higher or lower than 400 mg); **(D)** Infection Status (defined, following the article associated with each time-series, as whether the patient group were receiving treatment for a known infection or not); **(E)** Co-Administered Drugs (i.e. whether albendazole was delivered alone or in tandem with other drugs); and **(F)** Age Group, defined as to whether the average age of the patients was under 18 years (children) or ≥18 years (adults).

**Table 1. Regression outputs (p-values) relating pharmacokinetic properties to patients' characteristics.** Inferred parameters from the fitted pharmacokinetic model, specifically the median values of albendazole sulfoxide (AlbSO) systemic availability, albendazole sulfoxide clearance rate, albendazole sulfoxide $AUC_{400}$ and $C_{Max400}$ were regressed onto various patient group demographic and treatment metadata.

| | Systemic availability of albendazole sulfoxide | AlbSO clearance rate | $AUC_{400}$ (for standardised 400 mg dose) | $C_{Max400}$ (for standardised 400 mg dose) |
|---|---|---|---|---|
| **Fatty meal** | p = 0.03 (+14% in those receiving fatty meal) | p = 0.97 | p<0.01 (2.8x higher) | p = 0.001 (+314ng/ml) |
| **Sex (Ref = male)** | p = 0.38 | p = 0.78 | p = 0.33 | p = 0.19 |
| **Age group (Ref = adults)** | p = 0.03 (+15% in children) | p = 0.11 | p = 0.87 | p = 0.05 (+203ng/ml) |
| **Dose (mg)** | p = 0.11 | p = 0.13 | p = 0.29 | p = 0.08 |
| **Parasitic infection (Ref = none)** | p = 0.54 | p = 0.26 | p = 0.90 | p = 0.61 |
| **Neurocysticercosis** | p = 0.21 | p = 0.04 (+1 hour longer half-life) | p = 0.34 | p = 0.97 |
| **Echinococcosis** | p = 0.13 | p = 0.93 | p = 0.002 (2.9x higher) | p = 0.02 (+391ng/ml) |
| **Onchocerciasis** | p = 0.52 | p = 0.49 | p = 0.78 | p = 0.25 |
| **STH/LF/Giardiasis** | p = 0.50 | p = 0.62 | p = 0.17 | p = 0.17 |
| **Co-administered drugs (Ref = none)** | p = 0.15 | p = 0.52 | p = 0.55 | p = 0.28 |

In order to explore these relationships more formally, we carried out a multiple linear regression analysis to assess which of the factors in Fig 3 were statistically significantly associated with differences in these pharmacokinetic parameters. The results of this regression are displayed in Table 1. Receipt of a fatty meal prior to treatment increased the systemic availability of albendazole sulfoxide by 14% on average (p = 0.03) and resulted in a significantly higher peak plasma concentration ($C_{Max400}$ being 314 ng/ml higher in individuals receiving a fatty meal on average, p<0.001). Receiving a fatty meal prior to treatment was also associated with a 2.8-fold higher overall $AUC_{400}$ than in fasted individuals (p<0.01). We did not observe any significant differences in pharmacokinetic parameters that depended on sex, but observed an influence of age. Systemic availability was 15% higher in children than in adults (p = 0.03), and $C_{Max400}$ was 203ng/ml higher in children than in adults (p = 0.05). The dose received was not statistically significantly associated with differences in pharmacokinetic parameters.

Parasitic infection was associated with significant differences in pharmacokinetic parameters compared to uninfected individuals. Whilst we did not detect any significant differences when considering infection status as a binary indicator (i.e. whether an individual had a parasitic infection or not), stratifying the infected groups further by specific causal agent revealed significant associations between particular (manly cestode) infections. There was a significant association between neurocysticercosis and albendazole sulfoxide half-life (but the effect was marginal, with the estimate of the half-life longer by 1 hour in infected individuals, p = 0.04). There were also significant effects of echinococcosis on $C_{Max400}$ and $AUC_{400}$ (increased by 391ng/ml and 2.9 fold respectively, p<0.01 in both instances). We did not observe any significant association between the considered pharmacokinetic parameters and either (i) onchocerciasis; or (ii) soil-transmitted helminthiasis, LF, or giardiasis (considered as a single category due a paucity of data) and the considered pharmacokinetic parameters.

As a sensitivity analysis, we repeated the analyses described above controlling for the dose of albendazole received per kilogram of body weight (available only for a subset of the time-series due to a lack of complete information about participants' weight), rather than the raw amount (in mg, not standardised by body weight) given to an individual. All significant associations described above were retained when conducting this subset sensitivity analysis (see **Table B in** S1 File). Additionally, we observed a difference between age groups in the modelled

estimates of $k_{Albso}$, with the median clearance rate of albendazole sulfoxide 0.15 per hour higher than in adults, corresponding to a half-life of 12.4 hours in adults compared to only 7.6 hours in children under the age of 18 years (p = 0.01). We also observed significant associations between STH/LF/Giardiasis and both $C_{Max400}$ and $AUC_{400}$ (increased by 198ng/ml and 1.14-fold respectively, p = 0.04 and p = 0.02 respectively).

We did not detect a significant effect of co-administered drugs on albendazole's pharmacokinetics, though it is important to note that the heterogeneous array of drugs co-administered across the collated dataset, and the comparative paucity of time-series featuring each of the drugs precluded a stratified analysis of each drug separately (as was possible with infectious agent). This lack of data necessitated combining them into the binary category or yes/no co-administration. The corollary of this is that these analyses were not powered to reliably detect drug-drug interactions with albendazole (which are well documented in the literature).

## Discussion

Despite widespread usage, significant uncertainty surrounds the factors underlying variation in the pharmacokinetics of albendazole sulfoxide. Whilst other studies have previously examined these factors individually (e.g. [27,44,49,52,55] amongst others), a systematic analysis of different factors together remained outstanding. Integrating the results of a systematic review of the literature with a mathematical model of albendazole/albendazole sulfoxide pharmacokinetics, our work highlights the extensive inter-individual pharmacokinetic variation known to characterise albendazole sulfoxide pharmacokinetics, and the impact of a number of different factors in shaping the pharmacokinetic profile of albendazole sulfoxide in plasma following receipt of a single oral dose of albendazole.

In keeping with previous work [26,27,34,53,62], consumption of a fatty meal prior to receiving the dose was associated with increased systemic availability of albendazole sulfoxide (concomitantly elevating the $AUC$ and $C_{Max}$ values achieved) [28,29], a phenomenon thought to be attributed to changes in the drug's solubility (previously shown to be the rate-limiting step in albendazole's absorption [15]) when delivered alongside a fatty meal [62]. Whilst prior results from the literature have suggested (modest) differences between men and women in albendazole's pharmacokinetics (specifically with regards to the $AUC$ and $C_{Max}$ [24]), we did not observe any statistically significant differences here. However, important caveats to our results are that the lack of individual data in many cases precluded examination of men and women separately. Therefore, we constructed a crude proxy for comparison (between men and groups in which the group comprised mixtures of men and women), which may not have been powered to detect the (minor) differences previously reported [24]. Together, our results also suggest that different factors impact different pharmacokinetic properties of albendazole and albendazole sulfoxide. For example, whilst receipt of a fatty meal was associated with increases to systemic availability, $AUC_{400}$ and $C_{Max400}$ (in agreement with previous work exploring its effect [26,29]), age was significantly associated with systemic availability, albendazole sulfoxide half-life and $C_{Max400}$, with significant differences observed between adults and children even after controlling for differences in effective dosage due to differences in body weight.

Our analyses also suggested significant effects of parasitic infection on albendazole pharmacokinetics, with the exact impact dependent on the infection being considered. Previous work in sheep has highlighted that gastrointestinal nematode infection can influence the kinetics of albendazole and albendazole sulfoxide, leading to increased $AUC$ values [63]. Other work has noted increased rates of transformation from albendazole to albendazole sulfoxide in infected compared to uninfected sheep [64], though this has been inconsistently observed and the exact influence (or lack thereof) likely depends on the particular infecting parasite [65]. Work in

humans has suggested that the precise impact of infection depends on the interaction between the drug (particularly its absorption and elimination) and the infecting parasite's impact on the host. For example, whilst recent work comparing the pharmacokinetics of albendazole in uninfected and *Wuchereria bancrofti*-infected adults showed no differences [30], previous work exploring albendazole kinetics in 19 patients with echinococcosis (8 with *E. granulosus* and 11 with *E. multilocularis*) demonstrated delayed absorption and impaired elimination of the drug (with this latter effect contributing to increases in the *AUC* of albendazole sulfoxide, particularly in patients with extra-hepatic obstruction and cholestasis due to the disease) [38]. Consistently with these results, we observed a significant effect of echinococcosis on albendazole's pharmacokinetic parameters, with infection being associated with increases in both $AUC_{400}$ and $C_{Max400}$ of albendazole sulfoxide; likely driven by the same factors described previously. For neurocysticercosis, we observed alterations to the apparent half-life of albendazole sulfoxide though the effect was marginal. However, these results should be interpreted with caution. Sample sizes for each of the individual infections were small; for example, the largest was for echinococcosis with 14 time-series drawn from a total of four studies. The estimates presented here are therefore uncertain, and it is possible that study-specific variation not accounted for might explain the observed results. Relatedly, whilst we attempted to control for co-administered drugs, our ability to do this was limited (see below). It is, therefore, possible that the results presented here might be confounded by the receipt of treatment for an infection that is not described in the associated reference.

There are a number of limitations to the analyses presented here. Firstly, and perhaps most notably, the available data in the literature were highly heterogeneous, involving a diversity of treatment regimens (i.e. other co-administered drugs) and patients (i.e. characteristics), with data available at different levels of aggregation (i.e. individual vs. average profiles). This constraint limits the statistical power of our analyses to characterise the effects of different individual drugs on albendazole's/albendazole sulphoxide's pharmacokinetics. For example, whilst our binary indicator for co-administered drugs was not found to be significantly associated with any of the pharmacokinetic parameters explored here, numerous interactions between albendazole and other drugs such as cimetidine [14], azithromycin [66] and various anti-epileptic drugs [67] are well-documented in the literature. Relatedly, whilst we were able to explore the association of some factors with albendazole sulfoxide pharmacokinetic variability, information on other factors known to influence the pharmacokinetics of albendazole and albendazole sulfoxide (such as inter-individual variation in gastric pH which can significantly influence the drug's absorption given the pH dependence of its solubility [14–16]) was not present in the collated references, and so we were unable to quantify its impact on albendazole and albendazole sulfoxide pharmacokinetics.

In addition to these constraints posed by population-level data, the results presented here are limited in that they only describe the pharmacokinetics following treatment with a single dose of albendazole. This holds programmatic relevance given usage of albendazole in MDA programmes targeting STH [68] and/or LF [69] amongst others, but other treatment regimen exist, most notably the use of albendazole in dedicated clinical settings to treat individuals for diseases such as cysticercosis and echinococcosis. These regimens typically utilise multiple doses delivered over consecutive days. Previous results have indicated that albendazole appears to induce its own metabolism through induction of key enzymes in the liver [31], and that multiple doses given over sequential days can lead to changes in pharmacokinetic properties over the course of multiple dose regimens; specifically, reductions in the maximum plasma concentrations of albendazole sulfoxide reached following each dose [40]. However, the magnitude of this effect and the frequency of dosing required to elicit pharmacologically-relevant reductions in plasma concentrations remain far from clear and have, to date, been addressed in only a

limited number of studies. Exploration of this phenomenon and its consequences for anthelmintic treatment regimens using multiple doses of the drug would require both further clinical research and an extension of the mathematical model developed here, and likely represents an instructive avenue of future investigation. Indeed, whilst the overall modelling framework used here is similar to previously published models of albendazole sulfoxide pharmacokinetics (e.g. the use of explicit gut, peripheral and central compartments to capture relevant properties of albendazole's absorption and pharmacokinetics [70]), there are a number of additions to this framework that would likely provide new insight into albendazole sulfoxide's pharmacokinetics. These include the intestinal bioconversion of albendazole to albendazole sulfoxide known to occur [71], as well as infrequent but reported pharmacokinetic phenomena associated with albendazole treatment, such as biphasic pharmacokinetic profiles (thought possibly to be a product of inter-individual variation in frequency of gastric emptying affecting release of ingested albendazole into the gut, as well as other related characteristics [72]).

The availability of studies explicitly exploring the pharmacokinetics of drugs used to treat NTDs is limited [73]. Despite the limitations described above, our work begins to address this gap for albendazole. It suggests potential useful avenues for improvements to programmatic delivery of albendazole, and perhaps more importantly, highlights the existence of significant inter-individual variation in albendazole/albendazole sulphoxide pharmacokinetics. Whilst we provide insight into some of the factors underlying this variation, further quantification and exploration will be essential. This is particularly crucial for understanding and interpreting the results of studies exploring programmatic usage of the drug to treat parasitic infections. Having an understanding of the degree of inter-individual variation will be vital for establishing whether observed sub-optimal responses to the drug are due to parasitic factors (e.g. possible resistance potentially developed through cumulative exposure to treatment over multiple rounds [74]) or, instead, simply reflect a high degree of variation between individuals in total exposure to the drug (and hence anti-parasitic effect) that follows ingestion of the same oral dose. Together, these results support and underscore recent calls highlighting the need for the collection, collation and analysis of individual participant data (IPD) to generate robust evidence on efficacy and safety of anti-parasitic treatment regimens [75]. Given the increasing frequency with which albendazole is being utilised as part of community-based MDA programmes aimed at controlling a wide array of parasitic infections and NTDs, such an understanding would hold important public health relevance.

## Supporting information

**S1 File. Additional methods and results.**
(DOCX)

## Acknowledgments

We deeply thank Dr Annette C. Kuesel for her hugely insightful and helpful comments on earlier versions of this manuscript, which have materially improved and contributed to the work now presented here.

## Author Contributions

**Conceptualization:** Charles Whittaker, Cédric B. Chesnais, Sébastien D. S. Pion, Joseph Kamgno, Martin Walker, Maria-Gloria Basáñez, Michel Boussinesq.

**Data curation:** Charles Whittaker.

**Formal analysis:** Charles Whittaker.

**Investigation:** Charles Whittaker, Cédric B. Chesnais, Sébastien D. S. Pion, Joseph Kamgno, Martin Walker, Maria-Gloria Basáñez, Michel Boussinesq.

**Methodology:** Charles Whittaker, Maria-Gloria Basáñez, Michel Boussinesq.

**Project administration:** Maria-Gloria Basáñez, Michel Boussinesq.

**Resources:** Michel Boussinesq.

**Software:** Charles Whittaker.

**Supervision:** Maria-Gloria Basáñez, Michel Boussinesq.

**Visualization:** Charles Whittaker.

**Writing – original draft:** Charles Whittaker.

**Writing – review & editing:** Cédric B. Chesnais, Sébastien D. S. Pion, Joseph Kamgno, Martin Walker, Maria-Gloria Basáñez, Michel Boussinesq.

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
