## [Decision Letter · Decision Letter 0]

13 Jul 2022

Dear Mr Whittaker,

Thank you very much for submitting your manuscript "Factors associated with variation in single-dose albendazole pharmacokinetics: A systematic review and modelling analysis" for consideration at PLOS Neglected Tropical Diseases. As with all papers reviewed by the journal, your manuscript was reviewed by members of the editorial board and by several independent reviewers. In light of the reviews (below this email), we would like to invite the resubmission of a significantly-revised version that takes into account the reviewers' comments. 

We cannot make any decision about publication until we have seen the revised manuscript and your response to the reviewers' comments. Your revised manuscript is also likely to be sent to reviewers for further evaluation.

Sincerely,

Husain Poonawala

Academic Editor

Klaus Brehm

Section Editor

Reviewer's Responses to Questions

**Key Review Criteria Required for Acceptance?**

**Methods**

-Are the objectives of the study clearly articulated with a clear testable hypothesis stated?

-Is the study design appropriate to address the stated objectives?

-Is the population clearly described and appropriate for the hypothesis being tested?

-Is the sample size sufficient to ensure adequate power to address the hypothesis being tested?

-Were correct statistical analysis used to support conclusions?

-Are there concerns about ethical or regulatory requirements being met?

Reviewer #1: Did you do a PROSPERO registration? 

The authors should update their literature search as three years old! I had a quick search on Pubmed and there are a number of relevant studies, which should be included (e.g. the study by Hofmann has >250 participants, which is higher than all of the studies reported in your work. It is even mentioned as limitation in the discussion section….

Reviewer #2: The main objectives are clear and the inclusion criteria used to perform the systematic review are defined in the test

Authors developed a pharmacokinetic model within a a Bayesian framework adaptive Metropolis-Hastings based Markov Montecarlo which is commonly used for population models.

Reviewer #3: Please see comments below

**Results**

-Does the analysis presented match the analysis plan?

-Are the results clearly and completely presented?

-Are the figures (Tables, Images) of sufficient quality for clarity?

Reviewer #1: I would add the references of the included studies in the main paper not only in the appendix

Table 1: add soil-transmitted helminths

Reviewer #2: Figures and tables are clear and meet the quality requirements

Reviewer #3: See my speciific comments below

**Conclusions**

-Are the conclusions supported by the data presented?

-Are the limitations of analysis clearly described?

-Do the authors discuss how these data can be helpful to advance our understanding of the topic under study?

-Is public health relevance addressed?

Reviewer #1: Line 343: how does the model compare with a previously published pop PK model of albendazole?

Line 366: yes, this was described earlier. Please place your results in context with previous work. 

Line 369: I do not find this very interesting as this has been reported earlier (T. trichiura infection was for example found to be correlated with lower clearance of both albendazole metabolites)

Line 401: there is no paucity of data, just update your literature search….

Reviewer #2: Conclusions should state the most important outcomes of the manuscript. The interindividual variability has been recognized for a long time.

Reviewer #3: See comments below

**Editorial and Data Presentation Modifications?**

Reviewer #1: Abstract

This variation is thought to have important consequences for treatment success, please specify the disease that this was demonstrated

Line 37: STH infections were also shown to alter the PK of albendazole, hence this should be added here

Introduction

Line 92: and other confounding factors could be infection intensity or diagnostic method etc.. As for hookworm intestinal concentrations might be more relevant this is not an ideal example to mention that PK variations of albendazole have an influence on treatment success. This is pure speculation and has not been demonstrated

Reviewer #2: (No Response)

Reviewer #3: (No Response)

**Summary and General Comments**

Reviewer #1: Whittaker and colleagues have done a systematic review on albendazole and PK. This is a well written paper and an interesting analysis. I agree with the authors that limited data is available on pharmacokinetics on NTD drugs and this is a topic to be addressed. However, the study has an important limitation since the literature search is not up to date and therefore the authors missed recent studies, which will have an influence on their results.

Reviewer #2: - The main contribution of the manuscript is the effect of the parasitic infection on Albendazole Pharmacokinetics. Authors highlight the existence of a significant interindividual variation, however these results are very well known and have been previously reported in most of the articles, therefore conclusions should be oriented to the advance that authors found in order to understand the variability. 

- Authors state that the results correspond to albendazole or albendazole sulfoxide concentrations in blood. Most of the analytical assays use plasma for the quantification of the drug and its metabolites. Please explain

- In figure 2, differential equations assume that albendazole is absorbed in the gut, however it has been shown that the drug undergoes extensive intestinal bioconversion. If the equation were included in the model, would you expect a change in the results obtained? Please explain

- Line 243. Paragraph: Cmax value is 354 mg/ml higher or almost twice as high in individuals taking a fatty meal. is not clear. Albendazole sulfoxide levels are reported in micrograms/mL and not in mg/mL. Sentence should be clarified

- Different authors have shown that on of the main causes of interindividual variability is due to differences in gastric pH or intestinal metabolism. No mention about the gastric pH was included in the manuscript.

Reviewer #3: I´ve now completed the revision of the manuscript (PNTD-D-22-00636) entitled “Factors associated with variation in single-dose albendazole pharmacokinetics: A systematic review and modelling analysis”, which has been submitted for publication in PNTD. The manuscript describes the results obtained from a systematic review addressed to identify literature references containing temporally disaggregated data on the blood/plasma concentration profiles of albendazole and its main active metabolite albendazole suphoxide following a single oral dose. These data were then integrated into a mathematical modelling to infer key pharmacokinetic parameters and relate them to population features (age, sex, diet etc) which could affect drug pharmacokinetics and anthelmintic efficacy. This is a well written and presented article which may result of interest to the PNTD readership. It´s well known the key role of albendazole in MDA based STH control programs. Additionally, it is well established (since long time ago) that a number of host and drug formulation related factors may drastically affect albendazole/metabolites pharmacokinetic behavior and its resultant anthelmintic efficacy in different animal species and in man. Thus, the data obtained from the current systematic review is useful to highlight the relevance of the extensive inter-individual variation on albendazole pharmacokinetics which may explain often therapeutic failures against common STH species. Overall, this may be a useful contribution to the specific field since ratifies in a summarized format (using systematic review and modeling tools) well established principles in anthelmintic pharmacology. However, the article requires revision in many different aspects before it can be recommended for publication in PNTD. The following issues should be specifically addressed:

1.- The literature search was based on the Web of Science and PubMed databases until July 2019 using the keywords “albendazole” AND (treatment* OR dose* OR pharma* OR “half-life” OR “half life”) in order to identify references related to the main topics. Although a total of 5690 records were identified through this search process, only 32 studies containing temporally disaggregated data on drug/metabolite blood concentrations following treatment with a single dose were used to run the proposed model. Since research on albendazole and its performance on STH control programs is a very dynamic scientific field, I´d strongly suggest the authors to extend the systematic literature search until the current year. It sounds “deficient” and “incomplete” (outdated) to publish a literature search-based work whose cut-off line for incorporating data is 3 years old. 

2. There is a main issue to be corrected throughout the whole manuscript and fully address in a revised version of the article. Albendazole suffers a very efficient first-pass metabolism in the liver (and intestinal mucosa). Its sulphoxide and sulphone metabolites are the main albendazole related molecules recovered in the bloodstream. Only trace amounts of albendazole parent drug are occasionally recovered in the bloodstream in most of the animal species and man treated with a single oral dose. Although using modern chromatographic technology (HPLC MS/MS with very low detection sensitivity) it may be possible to detect “trace” albendazole concentrations in blood, any sound pharmacokinetic analysis can be performed for the parent drug. Traditionally, the estimations on albendazole systemic availability have been indirectly based on the availability of its main sulphoxide (active) and sulphone (inactive) metabolites. Thus, it´s unsound to refer to parent albendazole blood concentration profiles and even more inaccurate, to establish any conclusion on variations on its pharmacokinetic behavior. Could the authors check in how many of the 32 studies being incorporated was albendazole parent drug recovered in the bloodstream? 

On this reviewer´s opinion, the outcome of the current manuscript should be only focused on the identification of sources of variation to the pharmacokinetics of the active metabolite (albendazole sulphoxide) which can be recovered in the bloodstream and tissues of parasite location, and it´s the best “indicator” of albendazole absorption and systemic availability. The data shown in Figure 3 (Albendazole sulfoxide pharmacokinetic variability, stratified by patient and dosage features) is the most solid outcome of the performed work and should be a main focus of the revised version of the manuscript. To clarify and reinforce this main outcome of the reported systematic review and modeling (effect of fatty diet), the authors should revise recently published data on the impact of diet on albendazole sulphoxide systemic exposure in healthy volunteers treated with oral albendazole. The following references emerging from our work on those topics, as well as others available in the literature, should be illustrative of all the above described arguments. 

-Assessment of serum pharmacokinetics and urinary excretion of albendazole and its metabolites in human volunteers. Ceballos L, Krolewiecki A, Juárez M, Moreno L, Schaer F, Alvarez LI, Cimino R, Walson J, Lanusse CE. PLoS Negl Trop Dis. 2018 Jan 18;12(1):e0005945. doi: 10.1371/journal.pntd.0005945. eCollection 2018 Jan.

-Assessment of Diet-Related Changes on Albendazole Absorption, Systemic Exposure, and Pattern of Urinary Excretion in Treated Human Volunteers. Ceballos L, Nieves E, Juárez M, Aveldaño R, Travacio M, Martos J, Cimino R, Walson JL, Krolewiecki A, Lanusse C, Alvarez L. Antimicrob Agents Chemother. 2021 Aug 17;65(9):e0043221. doi: 10.1128/AAC.00432-21. Epub 2021 Aug 17.

3. In the light of above raised comments, it´s likely the authors may need to fully revise some of the ODEs (equations) governing the model applied here.

 4. There are some “terms” throughout the manuscript that should be revised to avoid confusion. The most of them are related to well established pharmacological terminology that sometimes is incorrectly used. The term “bioavailability” (absolute bioavailability) of a given drug can only be used when the plasma concentration profiles obtained by an extravascular route has been compared with those obtained after the intravenous administration. Systemic availability and/or systemic exposure are correct terms to refer to the amount of the administered drug reaching the bloodstream. Thus, is inaccurate to express the results and interpretation of your modeling as ….. " bioavailability of albendazole" and "AUC of albendazole sulfoxide". This is wrong, confusing and should be fully clarified. Otherwise, the interpretation and outcome of the systematic review and modeling is misleading. 

Additionally, pharmacokinetics and pharmacodynamics are well established definitions and extensively used terms in Pharmacology. The authors should be careful on how they use some terminology in the manuscript. Expressions such as “A number of factors are deemed to underlie this variation in pharmacokinetic dynamics (line 93), “we fitted a mathematical model of albendazole and albendazole sulfoxide’s dynamics in the blood following receipt of the dose that…(line 105)”, “….. our work highlights the extensive inter individual pharmacokinetic variation in albendazole’s dynamics (line 345). Please avoid use the term “dymanics” when you refer to pharmacokinetic issues/variables. This type of corrections should be applied both in the text, figure legends etc, to avoid confusion and misunderstanding. 

5. It has also been well established that the type drug formulation (preparation) may drastically affect the pattern of albendazole (and related benzimidazole compounds) dissolution at the stomach acidic pH and the resultant GI absorption and systemic availability of the sulphoxide metabolite. This is a critical issue for the interpretation of the work reported here. Are the authors sure that in all the 32 studies included in the analysis, albendazole was administered under the same tablet formulation? Was the same commercially available oral formulation used in all the trials? What about dosing? The studies were all performed with fixed albendazole doses (400 mg) or in some of them there was a dose calculation expressed in mg/kg body weight? This type of information should be provided if available or expressed as serious limitations on the reported data. 

6. Finally, the identification of sources of variation accounting for changes on albendazole dissolution, absorption and clinical performance can only be indirectly estimated by the systemic availability (exposure) and disposition kinetics of its main sulphoxide metabolite. The identification of host-related factors affecting the kinetic behavior of the main active albendazole metabolite, has been useful to establish strategies to improve albendazole anthelmintic performance avoiding common therapeutic failures both in Human and Veterinary Medicine. In many occasions, treatment failures are due to the incorrect use of the drug and not to genetic-based development of drug resistance as it´s very often attributed. For this reason, this reviewer strongly suggests the authors to correct the manuscript as recommended, considering this work is leaving a “useful take home message” on the need of further work to improve the use of albendazole (and other similar agents) in MDA campaigns for STH control.

PLOS authors have the option to publish the peer review history of their article (what does this mean?). If published, this will include your full peer review and any attached files.

Reviewer #1: No

Reviewer #2: No

Reviewer #3: No
---

## [Decision Letter · Decision Letter 1]

10 Oct 2022

Dear Mr Whittaker,

We are pleased to inform you that your manuscript 'Factors associated with variation in single-dose albendazole pharmacokinetics: A systematic review and modelling analysis' has been provisionally accepted for publication in PLOS Neglected Tropical Diseases.

Best regards,

Husain Poonawala

Academic Editor

Klaus Brehm

Section Editor

Reviewer's Responses to Questions

**Key Review Criteria Required for Acceptance?**

**Methods**

-Are the objectives of the study clearly articulated with a clear testable hypothesis stated?

-Is the study design appropriate to address the stated objectives?

-Is the population clearly described and appropriate for the hypothesis being tested?

-Is the sample size sufficient to ensure adequate power to address the hypothesis being tested?

-Were correct statistical analysis used to support conclusions?

-Are there concerns about ethical or regulatory requirements being met?

Reviewer #1: The authors have done an excellent revision and the manuscript is ready for publication

I would suggest only one change in the discussion:

....such as inter-individual variation in gastric pH which can significantly influence the drug’s absorption given the pH dependence of its solubility [14–16]) was not present in the collated references, and so we were unable to quantify its impact on albendazole and albendazole sulfoxide pharmacokinetics.

I would suggest rephrasing was not present in the collated references as I doubt that you expect the studies to undertake such an invasive procedure

Reviewer #3: (No Response)

**Results**

-Does the analysis presented match the analysis plan?

-Are the results clearly and completely presented?

-Are the figures (Tables, Images) of sufficient quality for clarity?

Reviewer #1: (No Response)

Reviewer #3: (No Response)

**Conclusions**

-Are the conclusions supported by the data presented?

-Are the limitations of analysis clearly described?

-Do the authors discuss how these data can be helpful to advance our understanding of the topic under study?

-Is public health relevance addressed?

Reviewer #1: (No Response)

Reviewer #3: (No Response)

**Editorial and Data Presentation Modifications?**

Reviewer #1: (No Response)

Reviewer #3: (No Response)

**Summary and General Comments**

Reviewer #1: (No Response)

Reviewer #3: I´ve now completed the revision of the revised version of the article (PNTD-D-22-00636R1) entitled "Factors associated with variation in single-dose albendazole pharmacokinetics: A systematic review and modelling analysis".

The manuscript has been fully revised according to the comments/suggestions raised by all (3) Reviewers. The revised version of the manuscript has substantially improved in comparison to the original submitted paper. The authors have accepted and incorporated the most of the suggested changes into the text of this revised version.

PLOS authors have the option to publish the peer review history of their article (what does this mean?). If published, this will include your full peer review and any attached files.

Reviewer #1: No

Reviewer #3: **Yes: **Prof Carlos Lanusse

---

## [Editor Report · Acceptance letter]

24 Oct 2022

Dear Mr Whittaker,

We are delighted to inform you that your manuscript, "Factors associated with variation in single-dose albendazole pharmacokinetics: A systematic review and modelling analysis," has been formally accepted for publication in PLOS Neglected Tropical Diseases.

Best regards,

Shaden Kamhawi

co-Editor-in-Chief

Paul Brindley

co-Editor-in-Chief
